# Highly Selective CO_2_ Hydrogenation to Methanol over Complex In/Co Catalysts: Effect of Polymer Frame

**DOI:** 10.3390/nano13232996

**Published:** 2023-11-22

**Authors:** Svetlana A. Sorokina, Nina V. Kuchkina, Stepan P. Mikhailov, Alexander V. Mikhalchenko, Alexey V. Bykov, Valentin Yu. Doluda, Lyudmila M. Bronstein, Zinaida B. Shifrina

**Affiliations:** 1A.N. Nesmeyanov Institute of Organoelement Compounds, Russian Academy of Sciences, 28 Vavilov St., 119991 Moscow, Russia; sorokinas@ineos.ac.ru (S.A.S.); kuchkina@ineos.ac.ru (N.V.K.); koal2702@yandex.ru (A.V.M.); 2Department of Biotechnology and Chemistry, Tver State Technical University, 22 A. Nikitina St., 170026 Tver, Russia; stefan.oblivion@mail.ru (S.P.M.); bykovav@yandex.ru (A.V.B.); doludav@yandex.ru (V.Y.D.); 3Department of Chemistry, Indiana University, 800 E. Kirkwood Av., Bloomington, IN 47405, USA

**Keywords:** CO_2_ hydrogenation, supported catalysts, methanol, indium oxide, polymer, cobalt, oxygen vacancies, basic sites

## Abstract

The growing demand for new energy sources governs the intensive research into CO_2_ hydrogenation to methanol, a valuable liquid fuel. Recently, indium-based catalysts have shown promise in this reaction, but they are plagued by shortcomings such as structural instability during the reaction and low selectivity. Here, we report a new strategy of controlling the selectivity and stability of bimetallic magnetically recoverable indium-based catalysts deposited onto a solid support. This was accomplished by the introduction of a structural promoter: a branched pyridylphenylene polymer (PPP). The selectivity of methanol formation for this catalyst reached 98.5%, while in the absence of PPP, the catalysts produced a large amount of methane, and the selectivity was about 70.2%. The methanol production rate was higher by a factor of twelve compared to that of a commercial Cu-based catalyst. Along with tuning selectivity, PPP allowed the catalyst to maintain a high stability, enhancing the CO_2_ sorption capacity and the protection of In against sintering and over-reduction. A careful evaluation of the structure–activity relationships allowed us to balance the catalyst composition with a high level of structural control, providing synergy between the support, magnetic constituent, catalytic species, and the stabilizing polymer layer. We also uncovered the role of each component in the ultimate methanol activity and selectivity.

## 1. Introduction

The depletion of fossil fuels has forced society to search for new alternative energy sources on the way to sustainable energy. In this regard, the concept of a methanol economy, proposed by G.A. Olah, has attracted significant attention [1]. Methanol can be synthesized from renewable sources, like biomass-derived syngas or atmospheric CO_2_, and it can be used as a liquid fuel in vehicles or polymer electrolyte fuel cells [2]. Moreover, considering the growing industry demand for methanol that is widely used in several industrial sectors, CO_2_ conversion to methanol could help to attenuate the harmful effects of CO_2_ emission on climate change [3].

The catalytic conversion of CO_2_ to methanol is an entropically unfavorable reaction and requires hard reaction conditions along with highly active catalysts to achieve a reasonable yield of methanol [3]. Methane, formic acid, and formaldehyde are also produced during the reaction, designating the reaction selectivity as one of the most difficult challenges facing the catalysts being developed [4,5]. Despite the number of the catalysts that have been designed during the last decade, which are primarily based on Cu, the methanol yield is still needs to be improved [6]. Another major shortcoming is the structural instability of the catalyst during the reaction, leading to its deactivation [7].

Recently, In_2_O_3_ was first theoretically predicted and then experimentally confirmed to possess a high activity in CO_2_ hydrogenation due to its abundant surface oxygen vacancies, which are believed to have a vital effect on the methanol-synthesizing activity [8,9,10]. Further studies have shown that the catalytic performance of In_2_O_3_ can be improved by its combination with other metals, e.g., Pd, Au, Pt, Ni, Ir, and Co [11,12,13,14,15,16,17,18]. The boost in activity is attributed to the formation of additional oxygen vacancies at the metal interface. Moreover, the interfacial sites facilitate CO_2_ activation and hydrogen spillover [19,20,21,22].

At the same time, the CO_2_ hydrogenation reaction is known to be structure-sensitive. In particular, metal loading and the size of the metal nanoparticles (NPs) have a great impact on the methanol productivity. For instance, an increase in Au loading leads to an increase in the methanol production rate for In_2_O_3_-based catalysts within the metal loading tested [12]. A similar trend was observed for Ir-modified systems [11]. The effect was attributed to the increase in the number of interfacial sites where the catalytic reaction takes place. The careful control of the indium content is also necessary. In [23], the authors revealed a boost in activity for a Cu-In-SiO_2_ catalyst with an increase in In loading. However, when the amount of In reached 5%, the activity dropped due to the dense coverage of the Cu with the InO_x_ species. Several studies have also hinted that a high dispersion of metal over the In_2_O_3_ support is an essential prerequisite for high methanol productivity [14,24]. Otherwise, the selectivity may significantly fall. The data indicate that the composition of In-based catalysts has to be carefully balanced and that the structure must be controlled at the atomic level to achieve an active system.

Despite the growing research interest in In_2_O_3_-based catalytic systems, they still suffer from several drawbacks. In particular, the structural instability during the CO_2_ hydrogenation reaction raises a major concern. Due to the low Hutting temperature, indium species are prone to sintering and aggregation under the reaction conditions [25,26,27]. Previous studies have revealed that indium doping with other metals enhances the stability of these catalysts [11,12,18]. Another problem is the reduction of surface atoms and the formation of In^0^/In_2_O_3-x_ species that deactivate the catalyst [10,28]. Recent studies have proposed that the deposition of indium onto a solid support may be an efficient strategy to overcome this limitation [10,27,29]. The authors showed that the introduction of ZrO_2_ into the catalytic system inhibited the over-reduction of indium. In this case, the formation energy of oxygen vacancies was higher than that of pristine In_2_O_3_, indicating that the interaction of In species with Zr prevented the reduction process. As a result, the oxygen vacancies that formed at the In–ZrO_2_ interface (In-Vo-Zr) were more stable and ensured a more active catalyst. At the same time, studies on supported indium catalysts are rather limited, and a deeper understanding of how the activity and selectivity of In-based catalysts may be regulated by the catalyst composition is required.

In our preceding work, we developed an approach to the deposition of a layer of hyperbranched pyridylphenylene polymer (PPP) onto a solid support by the Diels–Alder polycondensation of two branched monomers in the presence of silica gel [30]. The resulting composites proved their efficiency in the coordination of metal salts as well as in the stabilization of metal NPs synthesized by the wet impregnation technique [30,31]. Pyridines coordinate metals and redistribute the electron density, while the branched three-dimensional architecture holds the metal species in the pyridine vicinity and prevents the loss of the catalytic species. As a result, the activity and stability of PPP-based catalysts have been shown to surpass those of conventional heterogeneous catalysts many times [30,32,33,34,35].

Here, we constructed indium-based catalysts modified with cobalt oxide deposited onto a SiO_2_ support with embedded Fe_3_O_4_ magnetic nanoparticles and covered by a thin layer of PPP. Cobalt was chosen for its ability to activate hydrogen molecules and to enhance hydrogen spillover [36]. The magnetic NPs enabled the easy and cost-effective magnetic separation of the catalysts for their repeated use and can contribute to greener processes. Moreover, the role of Fe_3_O_4_ in the enhancement of the catalytic activity was also established. While SiO_2_ provided a support for the distribution of the catalytic species, the use of PPP was intended to prevent the over-reduction of In, aid in the stabilization of the metal particles, and protect against sintering. We further demonstrated that the presence of PPP was essential for the construction of the highly active catalytic system due to the manipulation of the methanol selectivity. The presented approach for the fabrication of these catalysts is a novel strategy that can be used to overcome the limitations of indium-based catalysts in CO_2_ hydrogenation, ensuring a high level of structural control. As a result, catalytic systems with an exceptional catalytic activity and selectivity can be obtained.

## 2. Materials and Methods

### 2.1. Catalyst Preparation

The synthesis of the support, namely, SiO_2_-Fe_3_O_4_-PPP, is described in [30]. Briefly, two monomers—a first generation of pyridylphenylene dendrimer bearing 6 ethylene groups and bis(cyclopentadienone) with two diene bonds—were allowed to adsorb onto the surface of mesoporous silica gel with pre-synthesized magnetic NPs. Afterward, a solvent (diphenyl ether) was added to the mixture, and the Diels–Alder polycondensation reaction was conducted over 10 h. The solid product was collected with an external magnet and thoroughly washed with organic solvents to eliminate unbound polymer molecules. The sample was dried in a vacuum oven and used for the deposition of the catalytic species.

Catalytic composites were synthesized by the wet impregnation method. For the synthesis of the 1.5In/0.3Co catalyst, 0.04 g of basic indium acetate dispersed in 2 mL of methylene chloride was added to 1.0 g of SiO_2_-Fe_3_O_4_-PPP. The mixture was allowed to evaporate at 30 °C and under constant stirring. After that, 0.028 g of cobalt acetylacetonate in 1.5 mL of methylene chloride was added to the mixture. The sample was dried in a vacuum oven and then heated in a furnace tube at 350 °C for 3 h with a heating rate of 2 °C·min^−1^. The amount of cobalt and indium precursors was adjusted depending on the desired metal loading. 

### 2.2. CO_2_ Hydrogenation Reaction

In a typical procedure, 50 mg of the catalyst and 15 mL of dodecane were placed in a 50 mL stainless steel reactor (PARR Instrument, Moline, IL, USA) equipped with a propeller stirrer and connected with a GS-MS Shimadzu 2010 (Shimadzu, Kyotocity, Japan) gas chromatomass spectrometer. Hydrogen was loaded into the system until the pressure reached 5 MPa, and the catalyst was reduced for 1 h at 250 °C. After that, the reactant gas (H_2_/CO_2_ = 4/1) was loaded into the system, and the stirring rate was set at 750 rpm. The reaction was conducted at 250 °C for 6 h. After cooling down the reactor to room temperature, the gas and liquid mixtures were separately analyzed with the chromatomass spectrometer. The gas phase was fed into the chromatographic system via a return pressure valve. The liquid was analyzed using calibration curves. The methanol production rate was calculated based on the following equation:(1)$$W=\frac{m(CH3OH)}{m(Me)\times \tau }$$
where *m* (*Me*) is the amount of catalytically active metal loaded into the catalytic reaction in kg, and *τ* is the reaction time in h.

For the recycle experiments, the catalyst was collected from the reaction mixture by means of an external magnet, washed with ethanol (80 mL) and acetone (80 mL), and dried in a vacuum oven. The catalytic reaction of the recycled catalyst was performed under similar reaction conditions as the first catalytic use.

### 2.3. Catalyst Characterization

The metal content of the catalysts was analyzed by X-ray fluorescence (XRF) measurements using a Zeiss Jena VRA-30 spectrometer (Carl Zeiss Jena, Oberkochen, Germany) equipped with a Mo anode, an LiF200 crystal analyzer, and an SD detector.

Nitrogen adsorption–desorption isotherms were recorded using a Quantochrome Nova 1200e (Quantochrome Instruments, Boynton Beach, FL, USA) analyzer at 77 K. All the samples were outgassed in a vacuum oven at 150 °C for 10 h before the measurements. The specific surface area was calculated based on the Brunauer–Emmet–Teller (BET) equation.

Transmission electron microscopy (TEM) and scanning transmission electron microscopy (STEM) images were recorded with an Osiris TEM/STEM (Thermo Fisher Scientific, Waltham, MA, USA) device equipped with a high-angle annular dark-field detector (HAADF) (Fischione, Export, PA, USA) and a Super X (ChemiSTEM, Bruker, Bradford County, FL, USA) X-ray energy dispersive spectrometer at an accelerating voltage of 200 kV. For the analysis, a Cu grid coated with a Lacey carbon film was placed into a suspension of the analyzed sample.

Powder X-ray diffraction (XRD) analysis was performed with a Proto AXRD Θ-2Θ diffractometer with a copper anode (Kα = 1.541874 Å, Ni-Kß filter) and a Dectris Mythen 1K 1D detector in the angular range of 2θ = 5–100°. The scanning step was set to be 0.02°, and the speed was 0.5°/min.

X-ray photoelectron spectroscopy (XPS) data were recorded with an Axis Ultra DLD (Kratos, Kyoto, Japan) spectrometer with monochromatic Al Kα radiation. The samples were degassed for 180 min before the analysis. The X-ray power was set to be 150 W. Survey spectra were obtained at an energy step of 1 eV with an analyzer pass energy of 160 eV. For the high-resolution spectra, these values were equal to 0.1 eV and 40 eV for the energy step and analyzer pass energy, respectively. The deconvolution was performed with the CasaXPS software 2.3.25.

The thermal desorption of carbon dioxide was studied using an Autochem gas chemisorption analyzer (Micrometrics, Norcross, GA, USA). Before the analysis, the catalysts were pretreated in a gas mixture (H_2_ (10 vol.%)/Ar (90 vol.%)) for 1 h at 300 °C. For the CO_2_ TPD measurements, 200 mg of the sample was placed into a U-shaped cell, purged with helium at a flow rate of 10 mL/min for one hour, and heated up to 470 °C. The temperature was maintained for one hour. After cooling to 105 °C, the carbon dioxide treatment was carried out at a flow rate of 10 mL/min for one hour. The sample was purged with helium and then heated to 470 °C. The signal of the desorbed carbon dioxide was recorded. The amount of desorbed carbon dioxide was estimated based on the calibration curve. 

The hydrogen temperature-programmed reduction (H_2_ TPR) of the samples was studied using an Autochem gas chemisorption analyzer (Micrometrics, Norcross, GA, USA). A total of 200 mg of the sample was loaded into a U-shaped cell and purged with helium at a flow rate of 10 mL/min for one hour. Helium was substituted with a gas mixture (H_2_ (10 vol.%)/Ar (90 vol.%)) for 30 min at the same flow rate, and the temperature was increased to 470 °C. The amount of hydrogen absorbed was determined using a calibration curve.

## 3. Results and Discussion

Here, we constructed In-based composites modified with cobalt with different loadings of metals. The catalytic species were supported on SiO_2_ containing pre-synthesized magnetic NPs and covered by a layer of hyperbranched aromatic polymer. It should be noted that the combination of In_2_O_3_ with cobalt may sufficiently enhance the catalytic activity due to the high dissociative ability of cobalt toward hydrogen. At the same time, along with speeding-up of the reaction rate, such a modification may turn the catalyst toward methane production without an appropriate structural adoption [37,38]. For example, in [38], an In_2_O_3_-based catalyst modified with cobalt showed a 39.8% selectivity toward methanol production. Gason et al. prepared an In/Co catalyst possessing an 80% selectivity [37]. Based on a previous report that highlighted the possibility of improving the reaction selectivity toward methanol formation by basic sites [39], we proposed using a branched pyridylphenylene polymer with weak basic sites as a structural promoter that could adjust the reaction pathway. Moreover, the branched three-dimensional architecture was expected to stabilize the catalytic nanoparticles and avoid the over-reduction and aggregation of In.

### 3.1. Catalytic Performance

Since the CO_2_ hydrogenation reaction is known to be structure-sensitive, particular attention was paid to balance all the structural constituents in order to afford the most active catalytic system. We comprehensively investigated the influence of the structural features imposed by the catalyst composition on the catalytic activity. For this purpose, the size, dispersion, and distribution of the catalytic species were adjusted by varying the metal loading. The role of the interfacial sites between the cobalt–indium–magnetic oxide that could boost the activity due to the formation of oxygen vacancies was assessed through the preparation of catalysts containing no cobalt or no magnetic oxide. The ratio of the metals was also tuned to find an optimum value that could provide the most efficient utilization of the catalytic species. And finally, to assess the influence of the polymer layer on the selectivity, activity, and reusability, catalysts without PPP were studied. A list of the prepared catalysts is presented in the Appendix A. The catalysts were denoted as xIn/yCo, where x and y are numeric letters corresponding to the nominal metal loading. All the catalysts contained a support—SiO_2_-Fe_3_O_4_-PPP—unless otherwise specified.

Figure 1 summarizes the results of the catalytic experiments. As a metric of the catalytic activity, the methanol production rate, calculated as the methanol amount (in grams) assigned to the metal loading per unit time (see experimental section), was used. Such an evaluation allowed us to estimate the intrinsic activity of the catalysts based on their molecular characteristics. All the experiments were conducted at 250 °C, since previous studies have shown that higher temperatures are unfavorable for In_2_O_3_ catalysts due to pronounced In over-reduction leading to a decrease in activity and selectivity [28].

The activity was compared with a Cu-ZnO-Al_2_O_3_ (Megamax) commercial catalyst that is normally used in the synthesis of methanol from syngas and is the most explored system in CO_2_ hydrogenation. As one can see, all the synthesized indium-based catalysts outperformed the conventional one. The introduction of Co promoted the reaction activity. Compared to the pristine monometallic In_2_O_3_ composite, all the bimetallic systems were more efficient. The best activity was achieved for the 1.5In/0.3Co composite, and it was 12-times higher than that of the commercial catalyst, which is an outstanding result. It should be noted that Co_2_O_3_ provides solely methane in the CO_2_ hydrogenation reaction with a good yield [36]. However, its combination with indium increased the methanol productivity. Interestingly, even though iron oxide does not possess any catalytic activity in CO_2_ hydrogenation, the magnetic NPs boosted the activity. The 1.5In/0.3Co catalyst without Fe_3_O_4_ yielded 657 g MeOH/kg Me·h (green column, Figure 1a), while the catalyst with the same metal loading containing magnetic NPs provided 952 g MeOH/kg Me·h. As one can see, the presence of the polymer had a decisive effect on the high methanol productivity. The composites without a polymer layer provided only 280 and 200 g MeOH/kg Me·h for the 1.5In/0.3Co and 3In/3Co catalysts, respectively (red columns, Figure 1a). The results demonstrated the effectiveness of the approach, where each element supplied a spike in activity due to the synergistic effect between the constituents.

According to Figure 1a, the metal loading had a major impact on the methanol productivity. There was an optimum metal content for both In and Co observed in this work. For the composites without Co, a 1.5% In content in the catalyst was found to be the most active. Thus, the 3In catalyst provided only 149 g MeOH/kg Me·h in comparison with 455 g MeOH/kg Me·h obtained for the 1.5In catalyst. However, the correlation between the metal loading and activity was non-linear, and a further decrease in the In content suppressed the activity. A similar trend was observed for the composites modified with Co. The influence of Co loading was studied with the use of a composite containing 1.5% In (Figure 1b). We found a volcano plot for the activity, with the In/Co ratio of 1.5/0.3% being the most effective. The results were contrary to previously published studies reporting a linear increase in activity with an increase in metal loading for In-based catalysts [11,12]. Our results indicated that there was a balance in the In/Co ratio that enabled the highest activity due to the most efficient utilization of the metals.

Along with methanol, methane formation was detected (Figure 1c). The amount of methane was strongly dependent on the metal loading. A better selectivity was observed for the composites with large amounts of metal. The selectivity reached up to 98.5% for the 3In/3Co composite. A decrease in the indium content to 1% reduced the selectivity to 89.7% for the 1In/0.5Co composite despite that overall yield of methanol being than that of the 3% composite. Nevertheless, the most active composite—1.5In/0.3Co—was characterized by an exceptional selectivity—95.4%. It is also worth noting that the observed selectivity values outmatched those reported earlier for In-based catalysts [40]. For example, an Au/In_2_O_3_ catalyst showed a selectivity of 67.8%, a Co/In_2_O_3_ catalyst showed 75%, and a Re/In_2_O_3_ catalyst showed 72.1% [12,17,41]. However, the largest effect on the selectivity was found to be provided by the polymer layer. The selectivity of methanol formation decreased to 70.2% for the 1.5In/0.3Co composite without the polymer layer, and the catalyst became more favored toward methane formation, undoubtedly indicating the prominent role of the polymer in supporting the high selectivity. A similar trend was observed for the higher metal loadings; however, it was less pronounced. 

The reusability of the catalysts was tested In four consecutive cycles. Figure 1d shows that 97 and 96% of the initial activity was retained after four cycles (24 h) for the 3In/3Co and 1.5In/0.3Co polymer-containing catalysts, respectively. The monometallic 1.5In catalyst was less stable and preserved 92% of the activity. The catalysts without a polymer layer demonstrated a more serious drop in activity—94 and 88% for the 3/3 and 1.5/0.3% composites, respectively. The results confirmed the excellent stability of the catalysts due to protective role of the polymer. 

### 3.2. Catalyst Characterization

An extensive investigation of the catalytic nanocomposites using a combination of characterization techniques was performed to examine the relationship between the structure and the catalytic properties. 

The bulk element compositions of the catalysts were determined by XRF and are presented in Appendix A. The observed metal content coincided with the experimental loading, and only minor deviations were found. We further focused our discussion on the 1.5In/0.3Co and 3In/3Co composites, which represented two examples of high and low metal loadings, in order to elucidate the influence of the structural properties governed by the metal loading on the catalytic behavior. To reveal the role of the polymer layer, the composite without PPP was comprehensively studied as well. 

The chemical composition of the catalysts was assessed by recording the EDX spectra (Appendix A), which confirmed the presence of the constituent elements and the purity of the catalysts. The textural properties of the composites were estimated by N_2_ adsorption–desorption measurements. The corresponding isotherms are presented in Appendix A. The specific surface areas were found to be 352, 319, and 213 m^2^/g for the 1.5In, 1.5In/0.3Co, and 3In/3Co catalysts, respectively, indicating a decrease in the surface area with the increase in the metal loading. 

The XRD pattern (Figure 2) revealed the presence of silica at 22° 2θ, a set of characteristic peaks of Fe_3_O_4_ (35.4°, 43.08°, 56.9°, 62.6° 2θ), and a set of reflections assigned to In_2_O_3_ (30.5°, 51.0°, 60.9° 2θ). Surprisingly, no diffraction peaks associated with Co were observed regardless of the Co loading. This could suggest that the the Co NPs were small and amorphous. The XRD pattern of the composite without PPP contained the similar set of reflections (Appendix A).

The morphology of the catalysts was established by STEM EDS mapping. The results showed a high dispersion of both the In and Co species over the SiO_2_-Fe_3_O_4_-PPP support for the composites with low metal loadings, as illustrated by the 1.5In/0.3Co composite in Figure 3a. The In and Co species were evenly distributed over all the composite, and no metal clusters were visible. The EDS maps of In, Co, and Fe were overlapped, indicating good intermixing. An increase in the metal loading worsened the distribution of the metal species. In the case of the 3In/3Co composite (Figure 3b), In aggregates could be well distinguished on the EDS maps. The Co species remained evenly distributed. Nevertheless, the EDS maps of In and Co matched the EDS maps of Si and Fe, indicating the location of the catalytic species inside the SiO_2_-Fe_3_O_4_-PPP space.

A careful investigation of the composites by HR TEM allowed us to detect the In_2_O_3_ NPs. The lattice fringes identified on HR TEM image of the 3In/3Co composite (Figure 4a) gave a d-spacing of 0.296 nm, corresponding to the (222) plane of In_2_O_3_. It should be noted that the size of NPs was difficult to determine due to their location inside the pores of the SiO_2_ with embedded Fe_3_O_4_ NPs. Additionally, the HR TEM results revealed the presence of Co_3_O_4_ NPs, as confirmed by the distance of 0.468 nm attributed to the (111) plane of Co_3_O_4_ (Figure 4b). Nevertheless, the number of distinct Co_3_O_4_ NPs was very small, since no diffraction peaks that could be attributed to these compounds were detected by XRD. In contrast, no In_2_O_3_ or Co_3_O_4_ NPs could be distinguished in HR TEM images of the 1.5In/0.3Co catalyst. These results confirmed a high dispersion of the metal species upon low metal loading.

The absence of PPP did not influence the distribution of the In and Co species over the SiO_2_-Fe_3_O_4_ support. The EDS maps of the 1.5In/0.3Co composite without PPP (Appendix A) were very similar to those of the catalyst containing PPP. Only a slight clustering of the In species was observed. Surprisingly, the presence of Fe_3_O_4_ was found to have a major impact on the distribution of the In species (Appendix A). The 1.5In/0.3Co composite without magnetic NPs demonstrated a clear aggregation of In in its silica pores. These results recognized Fe_3_O_4_ as a pool for the deposition of catalytic species, ensuring their uniform distribution over the support.

Summarizing the STEM EDS and TEM results, it is clear that the metal loadings had a significant impact on the dispersion of the catalytic species and thus influenced the catalytic activity. When the metal loading was low (1.5% for In and 0.1–0.9% for Co), both metal species were evenly distributed over the support. An increase in the metal loading lead to clustering of the NPs and the formation of aggregates. The important influence of the Fe_3_O_4_ NPs on the high dispersion of the In species was also established. Obviously, the CO_2_ hydrogenation reaction proceeds differently on different catalytic sites, e.g., highly dispersed or clustered. While highly dispersed In and Co species enable the existence of strong interactions between In, Co, and Fe (the so-called effect of strong metal–support interactions (SMSIs)), the aggregation of NPs weaken the efficiency of SMSIs [42]. At the same time, the effect of SMSIs is known to enhance CO_2_ hydrogenation to methanol due to the stabilization of the reaction intermediates [41,43,44]. Therefore, the presence of preciously dispersed catalytic species rather than clusters favors a superior catalytic activity. A similar effect was observed for a Re-modified In_2_O_3_-based catalyst [41].

To elucidate the reasons for the different losses of catalytic activity of the composites upon the recycle experiments, the catalysts were analyzed by STEM EDS mapping after the reaction. A loss in catalytic activity is usually associated with structural instability due to In sintering, which is the general case for In-based catalysts [26,28]. Our results showed that no In or Co aggregates could be identified in the STEM EDS maps of the 1.5In/0.3Co composite, nor were any changes observed for the 3In/3Co catalysts after the reaction (Appendix A). However, in the absence of PPP, a formation of In aggregates located at the edge of the composite was detected (Figure 5), which had not been found before the reaction. These results suggest that the deposition of a thin layer of coordinative aromatic polymer is an efficient strategy to overcome the structural instability and prevent In sintering. The coordination of PPP with metal species provides a reliable stabilization of catalytic sites and ensures superior stability. These results were well supported by the catalytic recycle experiments, where the composite without PPP—1.5In/0.3Co—showed a 12% loss of activity, while the catalyst with the same metal loadings containing PPP retained 96% of the initial methanol production rate (Figure 1d).

The surface chemical state of the catalysts was investigated by XPS. The survey spectra are presented in Appendix A, from which the presence of all the constituent elements can be identified. Figure 6a shows the HR XPS spectrum of In 3d for the 1.5In/0.3Co catalyst, which is deconvoluted into two peaks with binding energies of 445.4 and 452.9 eV for In 3d_5/2_ and In 3d_3/2_, respectively, corresponding to the In_2_O_3_ state [45]. An increase in In loading as well as the absence of a polymer layer did not affect the chemical state of In (Appendix A). The HR XPS spectrum of Co 2p of the 1.5In/0.3Co catalyst revealed that the Co was present in the oxidized state (Figure 6b). The peaks with binding energies of 781.5 and 796.9 eV were attributed to the Co 2p_3/2_ and Co 2p_1/2_ of Co_3_O_4_ [46]. Similar results were obtained for the other catalytic composites synthesized (Appendix A).

In addition, the HR XPS spectrum of Fe 2p was analyzed (Figure 6c). The deconvolution revealed the presence of two oxidizing states of Fe—Fe^2+^ and Fe^3+^-, with the main peak centered at 711.2 eV corresponding to the formation of Fe_3_O_4_ inside the silica pores [47].

To prove the coordination of PPP with the metal species, the HR XPS spectrum of N 1s was obtained. The spectrum was deconvoluted into three main peaks with binding energies of 399.0, 399.5, and 401.8 eV assigned to the nitrogen of the pyridine moieties, pyridine coordinated with metal, and quaternized pyridine, respectively. The results confirmed the coordination of PPP with the catalytic species.

Previous findings have postulated the tremendous effect of indium reduction on the catalytic activity, which occurs during the CO_2_ hydrogenation reaction. The surface reduction of In and the formation of In^0^/In_2_O_3−x_ was found to deactivate the catalyst [10,28]. The degree of deactivation was strongly correlated with the degree of In reduction. To elucidate the protective role of the polymer, the XPS spectra of the catalysts after the reaction were recorded to gain information about the changes in the chemical state of the catalytic species. Remarkably, no changes in the HR XPS spectra of In 3d and Co 2p of the 1.5In/0.3Co and 3In/3Co composites were observed (Appendix A). However, the formation of In^0^ in the composite without PPP was detected (Figure 6e). Thus, the deposition of PPP protects In against over-reduction and enables multiple catalytic cycles without any significant loss of activity. These results unambiguously indicate the necessity of a coordinative thermostable polymer layer to ensure the high stability and reusability of the catalyst. Interestingly, changes in the HR XPS spectra of Fe 2p were observed, where the ratio of Fe^2+^/Fe^3+^ increased, revealing an increase in the amount of Fe^2+^ species (See Appendix A for the deconvolution parameters and Appendix A for the XPS spectrum). This indicates that Fe_3_O_4_ acted as a reducible support, whose partial reduction during the reaction led to the formation of additional oxygen vacancies that boosted the activity [39,48,49]. These results explain the higher activity of the catalysts containing magnetic NPs that was observed in the catalytic experiments. These results are in line with previous findings recognizing the promotional role of interfacial metal-reducible metal oxide sites such as TiO_2_ and CeO_2_ in CO_2_ hydrogenation reactions [39,44,49]. While the oxides are considered to be inert, the emerging oxygen vacancies during the reductive treatment in the course of the reaction are believed to stabilize the reaction intermediates, provide more active sites, and improve the particle dispersion. Subsequently, this boosts the catalytic activity. At the same time, to the best of our knowledges the employment of Fe_3_O_4_ as a reducible support in CO_2_ hydrogenation reactions has not been reported earlier. Thus, magnetic NPs possess several functions in the considered process. They ensure easy and cost-effective separation for repeated use, enhance the particle dispersion, and provide more oxygen vacancies for more efficient catalysis.

To study the reducibility of the catalysts, H_2_ TPR experiments were carried out. As shown in Figure 7a, the reduction profile of the 1.5In monometallic catalyst contained a single broad peak with a maximum at 455 °C, which was assigned to the reduction of metal compounds (Fe_3_O_4_, Co_3_O_4_, and In_2_O_3_). The formation of bimetallic Co-containing composites led to the emergence of new peaks at lower temperatures. For example, the 3In/3Co catalyst was characterized by two small signals at 156 and 188 °C. These peaks were attributed to the surface reduction of In_2_O_3_ corresponding to the formation of oxygen vacancies [27]. Similar behavior was observed for the Co-modified systems without PPP (Appendix A) and without Fe_3_O_4_ (Appendix A). The most active catalyst, 1.5In/0.3Co, showed a larger broad peak centered at 169 °C, indicating that more oxygen vacancies were produced, which was consistent with the catalytic results. In addition, the existence of Co increased the hydrogen consumption, as was observed by the calculation of the amount of desorbed H_2_. It was found to be 0.34, 0.38, and 0.50 mmol/g for 1.5In, 3In/3Co, and 1.5In/0.3Co, respectively. Thus, these results confirmed the ability of Co to assist in hydrogen dissociation and spillover and to induce the formation of oxygen vacancies.

CO_2_ TPD measurements were conducted to further characterize the oxygen vacancies. Since oxygen vacancies are known to be produced under reductive conditions [50], as occurs during the reaction, the samples were pretreated under a flow of 10%H_2_ + 90%Ar for one hour at 300 °C. Figure 7b depicts the CO_2_ TPD curves of the 1.5In, 3In/3Co, and 1.5In/0.3Co catalysts. All the samples demonstrated a peak at 172–195 °C, which was assigned to the adsorption of CO_2_ onto the sites of hydrogen-induced oxygen vacancies [11,51]. The presence of a second peak at 355 °C could be seen for the 1.5In/0.3Co catalyst. This peak became broader and more intensive for the 3In/3Co composite. This peak was attributed to the sites of temperature-induced oxygen vacancies. These results imply the formation of new active sites for CO_2_ adsorption upon the formation of bimetallic Co-containing systems. It should be noted that the 1.5In/0.3Co catalyst without PPP had a much weaker intensity for the second signal (Appendix A). This suggests the participation of PPP in the creation of active sites for CO_2_ adsorption. We assume that the pyridine basic sites may have contributed to this process. The CO_2_ TPD profile of 1.5In/0.3Co without Fe_3_O_4_ is presented in Appendix A.

In summary, the developed strategy was shown to be highly efficient in the construction of stable, active, and selective In-based catalysts for CO_2_ hydrogenation to methanol. The deposition of PPP switched the catalyst toward methanol formation upon the modification of In with Co. We assume that the large hydrogen consumption and activation induced by the cobalt resulted in excessive hydrogen spillover onto the adsorbed reaction intermediates—formate species—which subsequently led to methane, as was observed for the catalysts without PPP. It was reported earlier that basic sites are able to improve the stability of formate intermediates, thus increasing the hydrogenation barrier [52]. We believe that tuning the selectivity by PPP was achieved through the stabilization of the intermediates by pyridines. At the same time, the PPP assisted in the CO_2_ adsorption, as was confirmed by the CO_2_ TPD measurements. Thus, the deposition of PPP allowed us to balance the adsorption and activation of the reactants, thus affording the optimum methanol production rate and selectivity. In addition to the structural adoption of the catalyst by means of setting the selectivity, the PPP also helped the In to resist sintering and over-reduction during the reaction according to the STEM EDS maps and XPS spectra of the used catalysts. This enabled a high stability and reusability.

While the selectivity and stability of the designed catalytic systems were regulated by the coordinative aromatic polymer, the activity was mainly influenced by the metal loading, which governed the dispersion of the catalytic species. We found that the high metal dispersion achieved at low loading favored a high methanol production rate. Moreover, an optimum metal loading and In/Co ratio were established. The data provided by a combination of characterization techniques are in line with the results of the catalytic performances and explain the superior activity of the 1.5In/0.3Co catalyst.

## 4. Conclusions

We demonstrated that the selectivity of a catalyst can be significantly improved by the introduction of a structural promoter such as a coordinative pyridine-containing hyperbranched polymer. In this work, we have shown that the deposition of PPP onto a catalytic nanocomposite can be used to tune the selectivity of methanol formation as well as to improve the stability of indium during catalysis. We observed the interplay between the SiO_2_, Fe_3_O_4_, PPP, and In/Co species, affording a superior methanol selectivity and production rate. The outstanding catalytic activity was a result of the balanced structural features providing the optimum CO_2_ and H_2_ adsorption and activation, hydrogen spillover, stability of the reaction intermediates, and number of oxygen vacancies. The SiO_2_ provided a support for the placement of the catalytic species and combined all the structural elements together. The magnetic NPs improved the dispersion of In_2_O_3_, boosted the activity due to the formation of oxygen vacancies, and ensure easy separation without the need for filtration, thus preventing the loss of the catalyst for multiple applications. The modification of the In with Co significantly enhanced the activity; however, it required an additional structural unit to direct the reaction pathway toward methanol production. We assume that the basic nature of PPP containing pyridine moieties adjusted the local environment of the catalytic sites, stabilized the reaction intermediates, and provided additional sites for CO_2_ adsorption. Moreover, the PPP enabled the preservation of the catalytic activity in the recycle experiments due to the stabilization of the In_2_O_3_ and the prevention of In reduction and aggregation. The developed approach allows one to overcome the major shortcomings of conventional In_2_O_3_-based catalysts and affords a new synthetic strategy that can be applied to the design of a wide range of catalysts suffering from structural instability.

## Figures and Tables

**Figure 1 nanomaterials-13-02996-f001:**
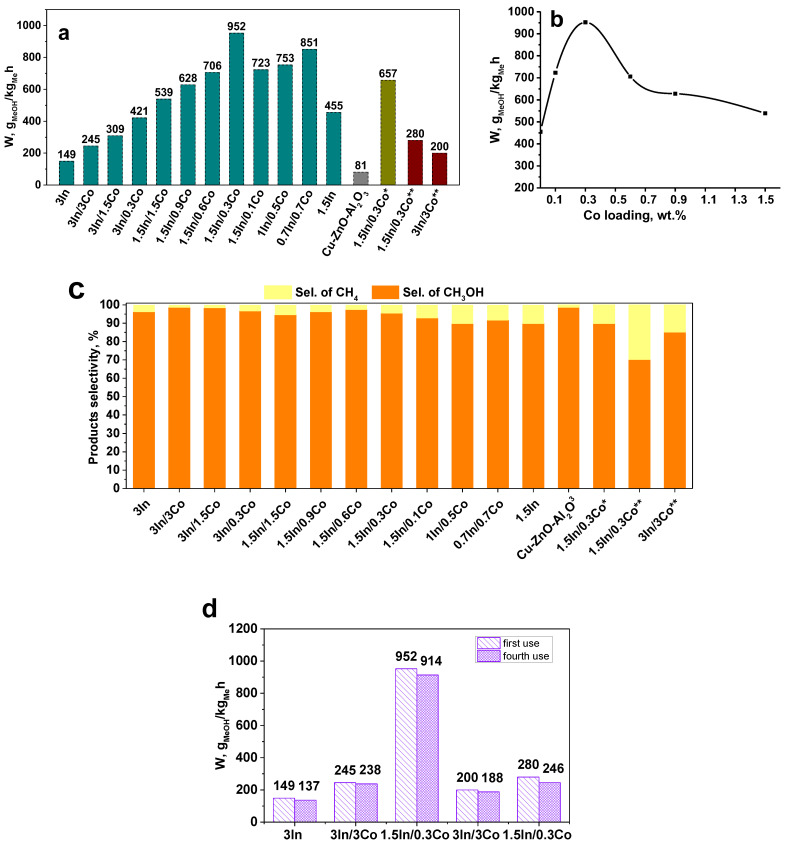
Results of the catalytic experiments. (**a**) Methanol production rate of the catalysts developed. (**b**) Effect of Co loading on the methanol production rate over 1.5% In loading. (**c**) Selectivity of catalysts in CO_2_ hydrogenation reaction. (**d**) Catalyst recycling experiments. Comparison of the methanol production rates obtained in the first and fourth use of the catalyst. * The composite does not contain magnetic NPs. ** The composite does not contain polymer layer of PPP.

**Figure 2 nanomaterials-13-02996-f002:**
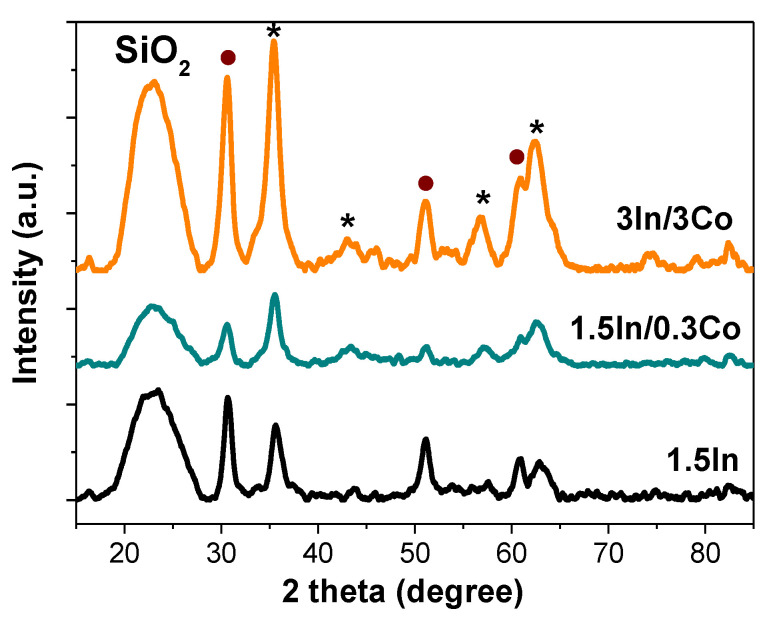
XRD patterns of 1.5In, 1.5In/0.3 Co, and 3In/3Co catalysts. The reflections assigned to In_2_O_3_ are marked with circles, and reflections assigned to Fe_3_O_4_ are marked with asterisks.

**Figure 3 nanomaterials-13-02996-f003:**
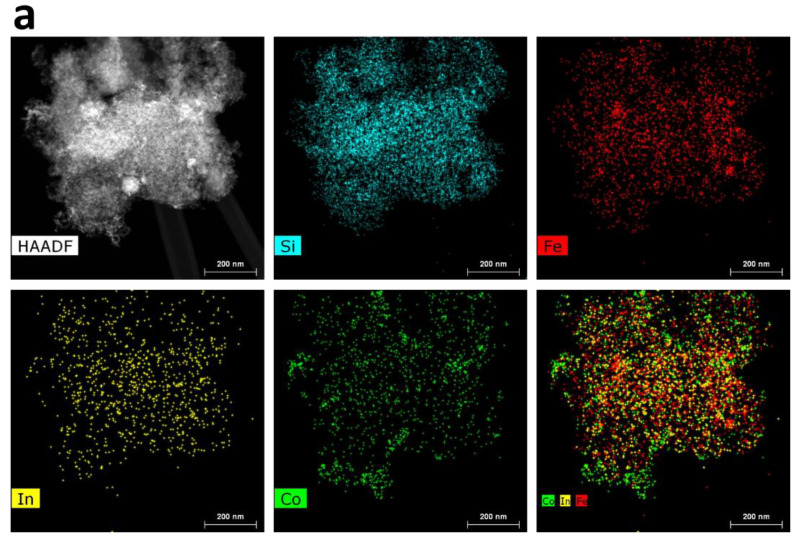
STEM dark-field image and EDS maps of Si, Fe, In, and Co and their superpositions of 1.5In/0.3Co (**a**) and 3In/3Co (**b**) catalysts.

**Figure 4 nanomaterials-13-02996-f004:**
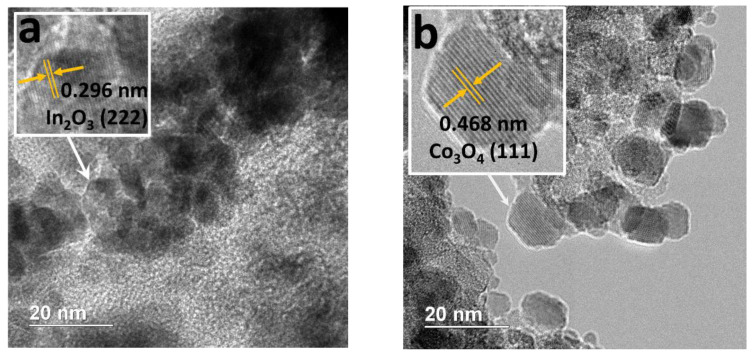
HR TEM images of 3In/3Co catalyst depicting In_2_O_3_ (**a**) and Co_3_O_4_ (**b**).

**Figure 5 nanomaterials-13-02996-f005:**
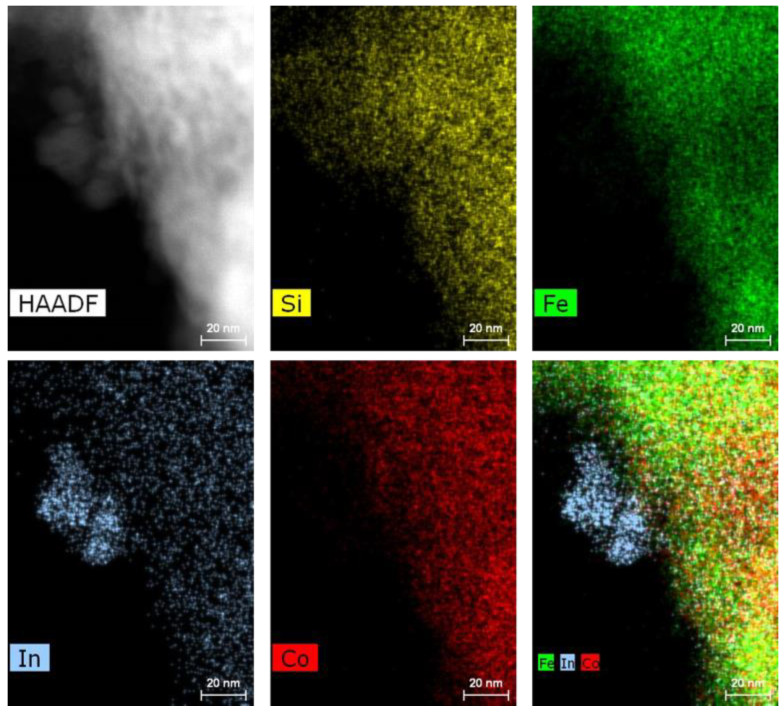
STEM dark-field image and EDS maps of Si, Fe, In, and Co and their superpositions of 1.5In/0.3Co without the polymer layer after the catalytic experiments.

**Figure 6 nanomaterials-13-02996-f006:**
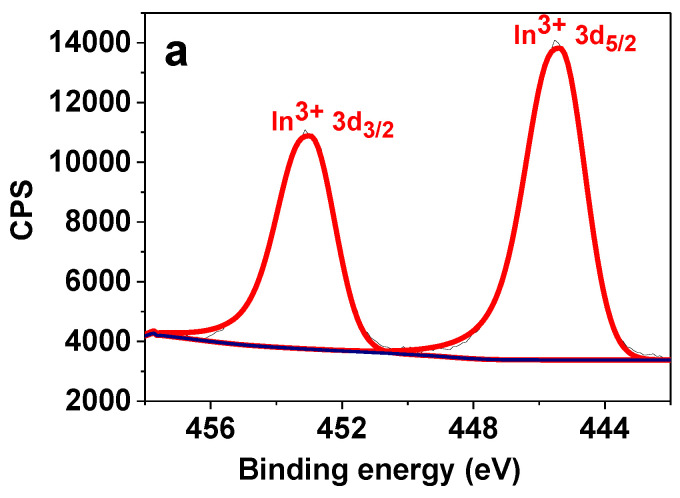
HR XPS spectra of the 1.5In/0.3Co catalyst in In 3d (**a**), Co 2p (**b**), Fe 2p (**c**), and N 1s (**d**) regions, and HR XPS spectrum of In 3d (**e**) of the 1.5In/0.3Co catalyst without PPP after the catalytic experiments.

**Figure 7 nanomaterials-13-02996-f007:**
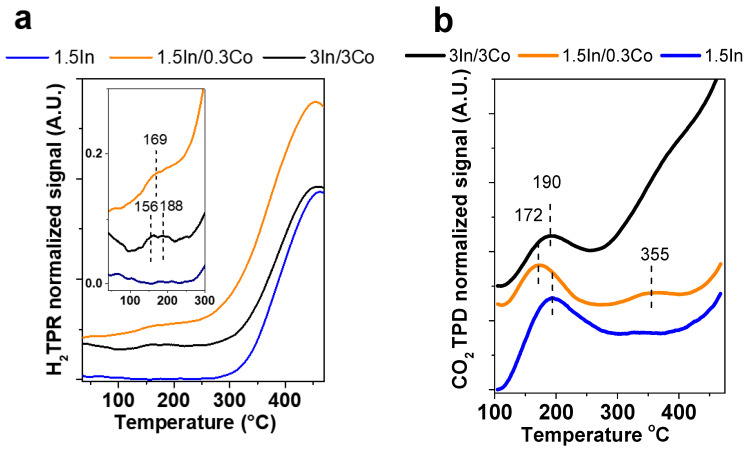
H_2_ TPR (**a**) and CO_2_ TPD (**b**) profiles of the 1.5In, 1.5In/0.3Co, and 3In/3Co catalysts. (**a**) contains built-in enlarge image of area 100–300 °C.

## Data Availability

The data presented in this study are available upon request from the corresponding author.

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
