# Peer review of "Highly Selective CO2 Hydrogenation to Methanol over Complex In/Co Catalysts: Effect of Polymer Frame"

_nanomaterials, 2023, doi:10.3390/nano13232996_

Round 1

Reviewer 1 Report

Comments and Suggestions for Authors

The authors highlighted the importance of PPP polymer in CO2 hydrogenation and tried to rationalize the better catalytic performance of 1.5In/0.3Co with polymer modification. Although the catalytic performance between the two catalysts is impressive, the fundamental understanding on the specific modification effects is very poor. This is mainly because the key reference catalyst, i.e., the 1.5In/0.3Co without PPP, was not characterized as the other samples. In this sense, it is impossible to gather the structural differences between the catalysts with and without PPP. Therefore, it is suggested the authors should fully characterize the key reference sample and correlate the catalytic performance with the catalyst properties.

Author Response

See the file

Reviewer 2 Report

Comments and Suggestions for Authors

In this work, the authors reported a new strategy of regulation of selectivity and stability of bimetallic magnetically recoverable indium-based catalysts deposited onto solid support by introduction of a structural promoter. The material exhibits high selectivity of CO2 hydrogenation to methanol. But the problems in the article still need to be improved. There some problems the authors have to fix before the paper can be published. The most significant ones are:

1)     Why different samples in XRD have similar peaking positions, but only changes in intensity?

2)     Keep the figures uniform in size in Figures 6 and 7.

3)     The characterizations of the material are not sufficient, and the existing characterizations are difficult to prove that the material successfully synthesized. Additional Infrared characterization can be added.

4)     Have the authors test SiO2-Fe3O4-PPP substrate without adding In/Co? And have they considered that the substrate has good catalytic activity for catalytic reactions?

5)     It is necessary to simplify the language in abstract so that it can express the core work of this paper more concisely.

6)     Some state-of-the-art literature is missing, such as, ACS Catalysis 2020, 10 (19), 11371; Nanomaterials 2023, 13(22), 2944; Carbon Neutralization 2022;1:4-5

Comments on the Quality of English Language

Can be improved

Author Response

See the file.

Round 2

Reviewer 1 Report

Comments and Suggestions for Authors

The authors have addressed the reviewer's concern. It is recommended for publication in the current form.